# Microrheometer for Biofluidic Analysis: Electronic Detection of the Fluid-Front Advancement

**DOI:** 10.3390/mi12060726

**Published:** 2021-06-20

**Authors:** Lourdes Méndez-Mora, Maria Cabello-Fusarés, Josep Ferré-Torres, Carla Riera-Llobet, Samantha López, Claudia Trejo-Soto, Tomas Alarcón, Aurora Hernandez-Machado

**Affiliations:** 1Department of Condensed Matter Physics, University of Barcelona (UB), 08028 Barcelona, Spain; josep.ferre@fmc.ub.edu (J.F.-T.); crierall7@alumnes.ub.edu (C.R.-L.); samantha.lopezm94@gmail.com (S.L.); a.hernandezmachado@ub.edu (A.H.-M.); 2Centre de Recerca Matemàtica (CRM), 08193 Bellaterra, Spain; cabello.maria1@gmail.com; 3Instituto de Física, Pontificia Universidad Católica de Valparaíso, Casilla 4059, Chile; claudia.trejo@pucv.cl; 4Catalan Institution for Research and Advanced Studies (ICREA), 08010 Barcelona, Spain; talarcon@crm.cat; 5Departament de Matemàtiques, Universitat Autónoma de Barcelona (UAB), 08193 Bellaterra, Spain; 6Institute of Nanoscience and Nanotechnology (IN2UB), Universitat de Barcelona (UB), 08028 Barcelona, Spain

**Keywords:** rheometer, microrheometer, rheology, hemorheology, viscosity, blood, plasma

## Abstract

The motivation for this study was to develop a microdevice for the precise rheological characterization of biofluids, especially blood. The method presented was based on the principles of rheometry and fluid mechanics at the microscale. Traditional rheometers require a considerable amount of space, are expensive, and require a large volume of sample. A mathematical model was developed that, combined with a proper experimental model, allowed us to characterize the viscosity of Newtonian and non-Newtonian fluids at different shear rates. The technology presented here is the basis of a point-of-care device capable of describing the nonlinear rheology of biofluids by the fluid/air interface front velocity characterization through a microchannel. The proposed microrheometer uses a small amount of sample to deliver fast and accurate results, without needing a large laboratory space. Blood samples from healthy donors at distinct hematocrit percentages were the non-Newtonian fluid selected for the study. Water and plasma were employed as testing Newtonian fluids for validation of the system. The viscosity results obtained for the Newtonian and non-Newtonian fluids were consistent with pertinent studies cited in this paper. In addition, the results achieved using the proposed method allowed distinguishing between blood samples with different characteristics.

## 1. Introduction

The study of the viscosity of biofluids, such as blood, has been used to understand their physiological mechanisms and biological functions [1]. Due to their complex properties, most biological fluids have a non-Newtonian behavior. While a Newtonian fluid shows a viscosity that is independent of the shear rate, non-Newtonian fluids’ viscosity shows a dependence on the applied shear rate.

As illustrated in Figure 1, blood exhibits shear-thinning behavior. Shear-thinning fluid viscosity is characterized by a monotonically decreasing dependence on the shear rate. Such behavior is closely related to the dynamics and interactions of red blood cells (RBCs), the main cellular component of blood. Blood viscosity variation as a function of shear rate due to rheological properties of RBCs is shown in Figure 1 [2,3,4,5]. RBCs have a biconcave disklike shape at rest; they aggregate and form rouleaux structures that can, reversibly and continuously, disaggregate to single-flowing discocytes for increasing shear rates. This change in microstructure implies a viscosity variation. When RBCs circulate through the vascular system, deformability and dynamics of RBCs account for a further decrease in blood viscosity. Furthermore, blood viscosity value strongly depends on other parameters such as temperature, whole blood cellular components (for example hematocrit value), and hemoglobin content [6].

The gold standard for viscosity measurements comprises a wide range of types of rheometers (instruments that measure the stress and deformation history of a material). The specific kind of rheometer that can only measure the steady shear viscosity function of a material is called a viscometer [8]. Rheometers may be differentiated on the type of flow they induce; i.e., drag flows and pressure-driven flows. Typical drag-flow rheometers are the coaxial cylinder Couette flow and coaxial cone-plate, while capillary rheometers are pressure-driven. The disadvantages of using this type of equipment include their high operation costs and the large size of samples they require. The rheometrical properties of complex fluids have been traditionally studied using such macroscopic devices.

In general, macroscopic rheometers provide less accuracy than their microfluidic counterparts. In contrast to classical methods, the study of rheology through microfluidic techniques is a widely used method to analyze the viscosity of complex fluids. Several microfluidic platforms have been developed to reach high confinement levels [9,10,11] (and thus, high precision), such as broader shear rate crystal microbalance (QCM) [12], laser-induced capillary wave [13], and the use of multiple microfluidic channels. This relatively new field has come up with some viscometers that measure viscosity as a function of shear rate and temperature, although most of them are not applied in biofluids. Novel microfluidic systems’ accuracy is equivalent to traditional methods, being also preferable for measuring complex viscosity behavior [14] and microflows [15]. A substantial number of viscometers for biofluid characterization exist [16,17,18]. However, they still present disadvantages in terms of space and sample amount required.

Some new applications use lab-on-a-chip structures from diverse materials aiming for miniaturization, such as glass and polydimethylsiloxane (PDMS) [19,20]. Microfluidic techniques that use shear as a mean to understand viscosity include pressure sensing, flow-rate sensing, surface tension, coflowing streams, and diffusion-based and velocimetry-based sensing [1]. The microrheometer proposed by Solomon [21] used a mobile phone camera to detect the fluid-front advancement through a microchannel. Srivastava [22], on the other hand, proposed a microrheometer in which the measurement of viscosity was based on monitoring with a camera the capillary pressure-driven movement of fluid sample, and shear rate varied with time. The device fabrication was not simple, and required much more time and specialized equipment and materials to produce, than our PDMS on glass microchannels. By contrast, the method of detection in our device is based on an array of electrodes that automatically detect the advancement of the fluid. By adding electronic sensing, we do not need further video processing and analysis. Additionally, our method has been demonstrated to measure viscosity of blood and differentiate between samples at distinct hematocrit levels.

The scale reduction allows for an increase in sensitivity and accuracy. Other factors such as hematocrit level, plasma, and coagulation could affect blood viscosity. On that matter, point-of-care devices such as thromboelastograph (TEG) (Haemoscope Corporation, Niles, IL, USA) and the ROTEM thromboelastometer (Pentapharm GmbH, Munich, Germany) [23,24] have led the research toward successful results for blood and plasma [25].

The technology presented here consists of a front microrheometer mainly composed of three parts: a microfluidic consumable that comprises a microchannel with an inlet and an outlet; the electronic setup, including pump, electronic detection, and data acquisition; and a mathematical model to properly convert obtained data into viscosity measurements. The electronic detection is performed with a sensor array (of metallic electrodes) located along the microchannel. This technology allows us to characterize the rheological properties of the fluid as a function of the fluid-front advancement, using a small amount of sample, reducing space required, and finally minimizing time and material consumption. The tested samples come into contact only with the walls of the microfluidic channel; therefore, there is neither a need for cleaning protocols, nor for the use of cleaning substances. Unlike other rheometers that require extensive cleaning of their components, the microfluidic chip is intended to be used as a consumable. The consumable is disposed of after each experiment. Thanks to the number of pairs of electrodes (24), situated at each side of the microchannel, it is possible to obtain a high number of data points, thus enhancing the accuracy of measurements. Finally, the integration of electronic detection into microfluidic devices improves the reliability of measurements.

The general goal of this study was to develop the method and model for a microrheometer capable of characterizing the viscosity of biofluids by analyzing the fluid-front advancement, and reducing the required sample’s amount, space, and time consumption. First, we characterized the viscosity of healthy blood samples at distinct hematocrit levels. The viscosity of blood obtained at $\dot{\gamma }=1$ was 12.2 mPa·s for the sample at 50% Ht; 10.67 mPa·s for 42% Ht; and 8.81 mPa·s for 35% Ht. These results were consistent with results obtained by other authors [26,27,28]. This constitutes a necessary benchmark to, in the near future, compare and detect rheological anomalies associated with hematological diseases, since the shape and biomechanical properties of RBCs can be altered in several diseases related to hematological disorders [29]. Second, we validated the new technology by comparing its results to data obtained using traditional equipment. We compared our viscosity results to results obtained using a rotational rheometer (Malvern Kinexus Pro+, Malvern Instruments Limited, Worcestershire, UK). The viscosity value obtained for blood at 42% Ht using this macrorheometer for comparison was 10.69 mPa·s.

## 2. Materials and Methods

### 2.1. Mathematical Model

The development of mathematical models describing the Newtonian and non-Newtonian characteristics of blood as a complex fluid [30,31] are very important for understanding viscosity behavior at the microscale, and determine how different blood conditions affect the viscosity values. Deformation of RBCs at high shear rates and the formation of aggregates in the form of rouleaux at low shear rates are essential aspects to keep in focus when developing models for blood [32,33].

The linear displacement of a fluid layer with respect to another in the time interval dt allows obtaining the rate of displacement (or velocity). If we consider a fluid flow moving into the x direction between two plates, the only nonzero velocity component is $v_{x}$, which varies in the z direction due to the interaction with the planes, being $\vec{v}=\left\{v_{x}\left(z\right),0,\;0\right\}$. The shear rate is the rate of change of velocity at which one layer of fluid passes over an adjacent layer. The shear produced between the layers depends on the z dimension, and is:(1)$$\dot{\gamma }\left(z\right)=\left({\mathrm{dv}}_{x}\left(z\right)\right)/\mathrm{dz}$$

An easy method to study fluid flow consists of the study of the fluid-front (fluid–air interface) advancement. In a microfluidic channel, the front velocity, $\dot{h}\left(t\right),$ is the change in position of the fluid, h(t), through time along the microfluidic channel. The term h(t) is defined as the average position of the front$\;h\left(\;t\;\right)=\frac{1}{n}\sum_{j=1}^{n}h_{j}\left(t\right)\;$, where $h_{j}\left(t\right)$ is the fluid-front position with respect to z, as shown in Figure 2. One of the advantages of working with the mean front velocity is that its value is independent of the z position. The theoretical model used assumes that velocity inside the microchannel behaves according to Darcy’s law. In past experiments [34], Trejo C. observed that this theoretical description of the velocity of the fluid inside the microchannel held for microchannels of 1 mm width and heights over 150 um. To test distinct aspect ratios, the authors considered microchannels of different heights: b = 300 μm, b = 200 μm, b = 150 μm, and b = 50 μm; width w = 1 mm; and length lc = 4 cm. For heights lower than 150 um the model is not longer valid.

By studying the fluid/air interface, the shear rate$\;\left(\dot{\gamma }\right)$ can be defined as the normalization of the mean front velocity, $\dot{h}\left(t\right),$ according to the microchannel height, b, as defined by Equation (1) (see Figure 2).
(2)$$\dot{\gamma }=\frac{\dot{h}}{b}$$

The viscosity of non-Newtonian fluids can be described using different models according to the characteristics of each fluid. The most commonly used are the Carreau–Yasuda and power-law models [35]. The power-law model, also known as the Oswaldt–De Waele fluid model, describes the viscosity of a fluid on dependence of its shear rate:(3)$$\eta \left(\dot{\gamma }\right)={m\dot{\gamma }}^{n-1}$$
where the prefactor *m* corresponds to the value of viscosity at $\dot{\gamma }=1$. The exponent *n* indicates the nature of the viscosity of the studied fluid. Depending on the value of *n*, the behavior of a fluid is characterized as Newtonian or non-Newtonian. Values of n<1 correspond to shear-thinning behavior, while n>1 corresponds to shear-thickening behavior. On the other hand, Newtonian fluids have a value of *n* = 1.

Using the Stokes equation, it is possible to calculate the flow through a known geometry as a function of the average velocity. In most cases, microfluidic systems are composed of pressure sources and microfluidic channels connected by cylindrical tubing, creating coupled fluidic systems. The flows passing through two coupled geometries are equivalent due to the mass conservation principle. The mathematical deductions concerning these calculations are detailed in Section A.1.

In a closed system composed of a pressure source, a fluid reservoir, and the coupling of tubing with a microchannel of rectangular cross-section, the pressure inside the rectangular microchannel, ΔP, is the summation of all the pressures in the system, as shown in Figure 3.

As properly deduced in Section A.2, the effective pressure ($P_{\mathrm{eff}}$) is defined as the pressure resulting from the combination of the pumping system pressure, the capillary pressure, and the hydrostatic pressure into the whole system shown in Figure 3 (see Equation (A25) deduced in Section A.2). Effective pressure $P_{\mathrm{eff}}$ has an algebraic dependence with $\dot{\gamma }$:(4)$$P_{\mathrm{eff}}=K\left(m,n\right){\dot{\gamma }}^{n\;}$$
where all the independent variables are grouped in K, which depends on the fluid properties (*m* and *n*) and the geometrical parameters of the system shown in Equation (4):(5)$$K\left(m,n\right)=m2l_{t}{\left(\frac{{\omega b}^{2}}{{\pi r}^{2}}\right)}^{n}\left(\frac{1}{r^{1+n}}\right){\left(\frac{1}{n}+3\right)}^{n}$$

According to the mass conservation principle, in our system, $Q_{c}=Q_{t}\to \omega b\dot{h}={\;\pi \;r}^{2}v_{t}$ (Section A.2). This is why we can write velocity inside the tubing as a function of the velocity measured inside the microchannel, $\dot{h}$:(6)$$v_{t}=\eta \frac{8l_{t}b\omega }{{\pi r}^{4}}\;\dot{h}$$

### 2.2. Experimental Model

#### 2.2.1. General Case: Non-Newtonian Fluids

We used a coupled system in which a tube was connected to a microchannel structure. The effective pressure, $P_{\mathrm{eff}}$, acting in the system (see Figure 3) caused fluid to flow through the coupled system. Through electronic detection, the mean front velocity was measured along the microfluidic channel. By calculating velocity values and the channel dimensions, shear-rate values were obtained. By plotting $P_{\mathrm{eff}}$ vs. shear rate, and fitting a curve for the relation between${\;P}_{\mathrm{eff}}$ and $\dot{\gamma }$, we obtained:(7)$$P_{\mathrm{eff}}={A\dot{\gamma }}^{n}$$

Terms A and *n* can be obtained from Equation (6). By equating experimental data in Equation (5) with the mathematical model in Equations (3) and (6), we could calculate the value of m as follows:(8)$$m=\frac{A}{2l_{t}{\left(\frac{{\omega b}^{2}}{{\pi r}^{2}}\right)}^{n}\left(\frac{1}{r^{1+n}}\right){\left(\frac{1}{n}+3\right)}^{n}}\;$$

Once *n* and *m* were obtained, the viscosity values of a fluid were calculated using the power-law model in Equation (2).

#### 2.2.2. Newtonian Fluids

A Newtonian fluid has a constant viscosity. In opposition to non-Newtonian fluids, it does not depend on the applied pressure. Analogously, as it has been done in the general case, the mathematical model for Newtonian fluids (n=1) is shown in Section A.3.

Pressure for the entire system is described in Equation (8), which has a linear dependence on the mean front velocity.
(9)$$P_{\mathrm{eff}}\left(m,n=1\right)=K\left(m,n=1\right)\dot{\gamma }$$

For the Newtonian case, K can also be expressed in terms of the geometry of the system and the experimentally obtained front velocity:(10)$$K\left(m,n=1\right)=m\frac{8{\omega b}^{2}l_{t}}{{\pi r}^{4}}$$

Along the same lines, the experimental data of the shear rate values obtained from applying a known pressure to a Newtonian fluid have the linear relation shown in Equation (10):(11)$$P_{\mathrm{eff}}=A\dot{\gamma }$$

Using the experimental results alongside the mathematical model, m can be calculated as:(12)$$m\left(n=1\right)=\frac{A}{\frac{8{\omega b}^{2}l_{t}}{{\pi r}^{4}}}$$

In Newtonian fluids for which *n* = 1, the power-law equation for viscosity in Equation (2) can be re-written as:(13)η=m

### 2.3. Experimental Method

The experiments were performed using microfluidic chips made of polydimethylsiloxane (PDMS) attached to glass substrates $\left(2.6\times 7.6{\;\mathrm{cm}}^{2}\right)$ by using plasma bonding; schematics of the final structures are shown in Figure 4a. The glass substrates had a printed pattern of 24 pairs of gold electrodes, which were in contact with the flowing fluid. The electrodes had a 350 μm separation in between, and were in the form of 4 groups of 6 pairs located along the length of the channel. The distance between each set was 8.5 mm (Figure 4b). The channel width was ω=1000 μm, the channel height was b=300 μm, and the length from inlet to outlet was $l_{c}=4\;\mathrm{cm}$. Commonly, PDMS, glass, and Cole Parmer Tygon tubing are employed due to their good biocompatibility, low cost, and high adaptability.

The experimental setup used to carry out the experiments was composed of a pressure pump (Dolomite Fluika) connected to a closed fluid reservoir. From the reservoir, Tygon tubing with an internal radius r=127 μm and length $l_{t}=20$ cm was connected to the microchannel. The pressure exerted on the fluid was set using a simple graphic interface, which controlled the pressure pump’s performance. A schematic representation of the experimental setup described is presented in Figure 5. Different pressures for the Fluika Pump were set to run the experiment, ranging from 500 Pa to 5000 Pa. Each electrode was connected via a pin array box to a National Instruments control card. As soon as the pressure began to be applied, electronic reading was activated by the controller of a myRIO National Instruments card. This tool communicated with the fluidic pump and the electronic reading pins through a computer. The sample fluid came out from the reservoir at the set pressure and went directly into the channel. As the fluid came into contact with the electrodes, the time required for the fluid front to reach each electrode pair inside the microfluidic channel was obtained. Using the time data and the geometry of the microfluidic system, it was possible to calculate the fluid-front velocity between electrode pairs through the microchannel.

Ten samples of 500 μL from each different test fluid were used during this study: (fluid standard MGVS60, deionized water (DI water), plasma, and blood from healthy donors). We selected MGVS60, a viscosity standard tested in strict accordance with ASTM D2162, the primary method for viscosity standards calibration that is typically used for the calibration and verification of viscosity-measuring equipment. Its viscosity is 6.00 mPa·s at 24 °C. Deionized water was selected because its viscosity is constant and known: η=1.01 mPa·s [36]. Blood samples and their respective plasma were provided in 10 mL tubes by Banc de Sang i Teixits (Barcelona, Spain), from 10 anonymous healthy donors. The use of these samples was authorized by the Bioethics Committee of the University of Barcelona (IRB 00003099).

To obtain the plasma samples, we centrifuged whole blood samples at 2500 rpm for 5 min. The plasma that rested at the top was collected with a pipette. Samples were prepared at three different hematocrit concentrations, Ht (%) of 50%, 42%, and 35%. These were prepared by taking plasma previously separated from whole blood and then adding to it the desired volume of RBCs corresponding to each hematocrit percentage. The hematocrit selected for further comparison to a benchtop device was 42% Ht, consistent with the levels found in healthy adults. All the tests were performed at room temperature (25 °C) and, for the case of blood, within two days of extraction, to avoid cell damage from aging [28]. At the beginning of each experiment, the DI water was evaluated to guarantee that the setup was properly calibrated.

## 3. Experimental Results and Discussion

For a correct rheological characterization of the fluid, it was necessary to properly determine the pressures inside the system (Figure 3). The pressure set for the pumping source $\left(P_{p}\right)$ was automatically recorded. Considering that a sample of 500 μL inside a reservoir supposes a height, H=5 mm, the contribution generated by the hydrostatic pressure $\left(P_{H}=\rho \mathrm{gH}\right)$ can be obtained. Parameters for the different tested fluids at room temperature can be found in Table 1. The material used for the microchannel was hydrophobic, meaning that it opposed the fluid-front advancement. Fluid-front images at different pressures were obtained by using a camera and an inverted microscope. The contact angle, θ, was extracted optically via Image J free software. The surface tension (τ) of the standard fluid was experimentally measured with a capillary tube with an internal diameter of 1.15 mm [37,38]. By using Equation (A20) deduced in Section A.2, it was possible to calculate $P_{\mathrm{cap}}\;$for a rectangular cross-section channel at each measured pressure by analyzing the fluid-front contact angle. The contact angle also depended on the fluid being analyzed and the total pressure applied. The curvature of the fluid–air interface was expected to be almost constant for water, standard MGVS60, and plasma; however, it changed as the pressure was increased when measuring blood (Table 2).

When studying blood-front behavior, dependence between the applied pressure and the dynamic contact angle measured was observed. Hence, $P_{\mathrm{cap}}$ depended on the applied pressure, $P_{p}$. To obtain the contact angle of blood inside our microfluidic channel, we conducted a separate experiment for blood at 42% Ht. By using a microscope (Optika XDS-3, Optika SRL, Bergamo, Italy) and a high-speed camera (Photron Fastcam SA3, Photron Limited, Tokyo, Japan), we captured images of the fluid front at different set pressures, $P_{p}$. We measured the contact angle using the angle-measurement tool in Image J, as depicted in Figure A2. Calculation of $P_{\mathrm{cap}}$ was done as per Section A.2. The obtained contact angle when changing the applied pressure and thus, the capillary pressure obtained can be seen in Table 2. The contact angles of blood used in this paper have been considered independent of the Ht (%) value. 

To validate our setup and the use of electronic detection as a proper method to determine viscosity for Newtonian fluids, two different strategies were followed: first, a Newtonian standard fluid and DI water with reported viscosity value were evaluated. Second, the obtained results with our technology were compared to the results obtained with a commercial macrorheometer, the Malvern Kinexus Pro+ (Malvern Instruments Limited, Worcestershire, United Kingdom).

From the data obtained (applied pressure and time), knowing the device geometry, resistances, and the pressures added to the system, it was possible to plot the effective pressure as a function of shear rate for each sample.

A linear fit of the data obtained when plotting the applied pressure with an external pressure pump $\left(P_{p}\right)$ as a function of $\dot{h}$ or $\dot{\gamma }$ is shown in Figure 6. It was used to determine $P_{\mathrm{eff}}$ for Newtonian fluids, as detailed in Equation (A33), as:(14)$$P_{\mathrm{eff}}\left(n=1\right)=m\frac{8{\omega \mathrm{bl}}_{t}}{{\pi r}^{4}}\dot{h}=m\frac{8{\omega b}^{2}l_{t}}{{\pi r}^{4}}\dot{\gamma }$$

As shown in Figure 6 and detailed in Table 3, in both cases, *n* was very close to 1, which indicated that the fluid was Newtonian, compatible with the behavior expected for both.

From these data, as previously explained, the viscosity could be obtained by using Equation (13). These results were compared to those obtained with a commercial macrorheometer (see Figure 7).

Water has been found to exhibit a constant viscosity of η ≈ 1.002 mPa·s [39]. The same results were obtained with both rheometers (differing from each other by 3%). The viscosity obtained with the standard MGVS60 was around η≈5.55 mPa·s using the microrheometer, while the macrorheometer delivered a viscosity of 5.8 mPa·s. The expected value for this fluid was 6.00 mPa·s.

These results validated that the accuracy of the new electronic sensing method was adequate (equivalent to standard macrorheometer measures) regarding the measurement of viscosity in an automatic, fast, and effective way.

Once the measurement system was validated for Newtonian fluids with known characteristics, the experiment was repeated with plasma and blood at different % Ht values. The results are shown in Figure 8.

The behavior of the shear rate as P_eff_ applied to the fluid variance is shown in Figure 8. Non-Newtonian fluids (in this case, blood as a shear-thinning fluid) showed a nonlinear relation between the pressure and their shear-rate response, while plasma showed a Newtonian behavior. Using the experimentally obtained values alongside Equations (6) and (7) for the non-Newtonian case, and Equations (10) and (11) for the Newtonian cases, the value of K could be calculated. Then, *m* and *n* were obtained. The values for each are shown in Table 4 (see references in Appendix B); water and MGVS60 values are also included.

With the effective pressure, and after obtaining m and n (Table 4), viscosity plots were obtained. Viscosity was calculated using Equation (2) (for non-Newtonian fluids: blood at different Ht (%). levels) and Equations (11) and (12) (for Newtonian fluids: plasma).

As depicted in Figure 9, plasma exhibited Newtonian behavior with a constant viscosity, between 1.50–1.81 mPa·s [40]. Blood exhibited shear-thinning behavior, as its viscosity decreased as the shear increased. Note that the viscosity of blood at 42% Ht (see Figure 9) varied from 12 to 7 mPa·s within the $\dot{\gamma }$ range evaluated, with a value of viscosity 10.67 mPa·s at $\dot{\gamma }=1\;s^{-1};$ blood at 50% Ht had a viscosity of 12.2 mPa·s; and blood at 35% Ht had the lowest viscosity of the three tested hematocrit levels, 8.8 mPa·s. At a constant shear rate, the viscosity of blood increased as the hematocrit grew. This also demonstrated that our new device could be used to distinguish between different hematocrit levels. The hematocrit selected for further comparison to a benchtop device was 42%, consistent with the levels found in healthy adults. The viscosity obtained by this method was 10.69 mPa·s. for blood at 42% Ht. This can be compared to results also obtained under room-temperature conditions by consulted authors, who situated the viscosity of blood at $\dot{\gamma }=1\;s^{-1}\;$between 9–10 mPa·s [40]. Dispersion in the measurements at the low shear-rate values observed in Figure 9 would be reduced by increasing the number of samples studied. With a sufficiently large N, a general curve and validity intervals for healthy blood could be obtained, which would indicate what the behavior of the viscosity must be as a function of the shear rate in a healthy patient (at a constant hematocrit value).

Regarding the rheological properties of RBCs, blood showed higher resistance to flow at low $\dot{\gamma }$ values. In contrast, at higher values of $\dot{\gamma }$, blood showed a more Newtonian behavior, which was reflected in the nearly flat region on the viscosity vs. shear rate plot. This distinct shear-thinning curve behavior could only be properly observed when shear rates as low as $1{\;s}^{-1}$ were reached. For this reason, it was important to choose a range of pressure that allowed for this $\dot{\gamma }$ regime to be available. Residual fit for both methods was a useful way to determine the accuracy differences between our technology and a commercial rheometer (see Figure 10).

Residuals, R, for the micro- and macrorheometer were calculated as the difference between experimentally obtained viscosity, η, and predicted viscosity, $\eta_{\mathrm{Predicted}}$ (Equation (14)).
(15)$$R=\eta -\eta_{\mathrm{Predicted}}$$

The theoretically predicted viscosity of blood, $\eta_{\mathrm{Predicted}}$, for both methods was obtained by calculating viscosity at each shear rate, using the fit of viscosity vs. shear rate from plots in Figure 9. For the micro- and macrorheometer, the calculation was made using Equations (15) and (16) respectively:(16)$$\eta_{\mathrm{Predicted}\;\mathrm{micro}}=10.674{\dot{\gamma }}^{-0.148}$$
(17)$$\eta_{\mathrm{Predicted}\;\mathrm{macro}}=10.687{\dot{\gamma }}^{-0.146}$$

A good correlation was observed when plotting experimentally obtained values for viscosity and theoretically predicted values using the power-law model shown in Figure 10a, thus confirming that the experimentally obtained data was close enough to theoretically expected values. In addition, residuals’ plotted vs. predicted values in Figure 10b show that the macrorheometer residuals were larger than those obtained with our microrheometer. The accuracy of the prediction for the macrorheometer in this experiment was lower than the accuracy of our microrheometer by three orders of magnitude. Residuals for our microrheometer were closer to zero. At these low shear rates, from 1 to 25 $s^{-1}$, our technology performed with higher accuracy. This could be explained by the fact that blood is a complex fluid of the shear-thinning type. We decided to work on this $\dot{\gamma }$ window to obtain the nonlinear viscosity of blood and to avoid hemolysis. Our microrheometer had the advantage of being able to operate under low shear conditions with high stability. This makes it suitable for applications such as the analysis of blood, where low-shear-rate performance is necessary. At the shear-rate values required to observe the non-Newtonian behavior of blood, the microrheometer results presented herein had a smaller deviation. This can be attributed to the high stability at low shear rates. These low shear rates could be reached thanks to the size reduction achieved by using a microchannel. Moreover, a benchtop rheometer for laboratory use also requires a rather large sample of blood. For the macrorheometer experiments, the sample amount was 1.19 mL for each run. The microrheometer required samples of approximately 250 μL; this constituted 21% of the sample required for the benchtop device.

## 4. Conclusions

The technology presented here was based on the detection of the advancement of the fluid front using electrodes printed underneath a microfluidic channel. The accuracy of the method was tested by measuring the viscosities of plasma, DI water, and a commercial viscosity standard. Experiments to measure human blood viscosity also were performed. Unlike other microfluidic capillary viscometers, the microrheometer presented here was based on the detection of the fluid/air interface as it advanced through a microchannel. Through the experimental method described, it was possible to produce a series of shear rates that constituted a proper range to observe how the viscosity of non-Newtonian fluid changed as the applied pressure increased. Effective pressure, the pressure acting on the fluid is composed of all the interactions between the pressures acting in the system.

The characterization of the non-Newtonian behavior was obtained by applying the power-law model in Equation (2). The experimentally obtained results included a shear-rate window range of less than $1{\;s}^{-1}\;$up to $100{\;s}^{-1}\;$for both Newtonian and non-Newtonian fluids. The values obtained for viscosity—1.05 mPa·s for water and 1.8 mPa·s for plasma—fell in line with the values found in the literature [36,39]. Alternatively, we used the calibration standard MGVS60 to test the accuracy of the setup and obtained a viscosity value of 5.88 mPa·s, showing a difference of less than 2% from the target viscosity indicated by the manufacturer at 24 °C. On the other hand, blood results showed shear thinning behavior, due to the presence of mainly red blood cells, under normal morphological conditions [2]. This confirmed that the experimental conditions were adequate to observe the shear-thinning behavior and the action of red blood cells on the viscosity of the whole fluid. The viscosity of blood at $\dot{\gamma }=1$ was 12.2 mPa·s for the sample at 50% Ht; 10.67 mPa·s for 42% Ht; and 8.81 mPa·s for 35% Ht. In addition, the viscosity value obtained for blood at 42% Ht using a macrorheometer for comparison was 10.69 mPa·s. As expected, as the hematocrit level increased, so did the value of viscosity. We therefore concluded that the new method proposed in this article is a reliable and accurate device that exhibits a degree of sensitivity capable of differentiating between these distinct hematocrit percentages. In addition, we measured the viscosity of DI water, MGVS60, and blood using a macroscopic rheometer for comparison purposes. Focusing on non-Newtonian fluids, we compared the behavior of the residuals of blood viscosity obtained with both rheometers.

Throughout this study, it has been demonstrated that the microrheometer using electronic detection of the fluid front can deliver viscosity results comparable to those obtained using traditional equipment, while saving time and reducing the sample size and space required. At the same time, this technique permits a better control at low shear rates for shear-thinning fluids, such as blood. The development of this experimental and theoretical method for blood will be useful in the near future to determine the presence of RBC abnormalities through the rheological characterization of blood. It could be useful for medical diagnosis in the study of diseases associated with changes in blood viscosity, operating in a wide range of shear rates [41].

## Figures and Tables

**Figure 1 micromachines-12-00726-f001:**
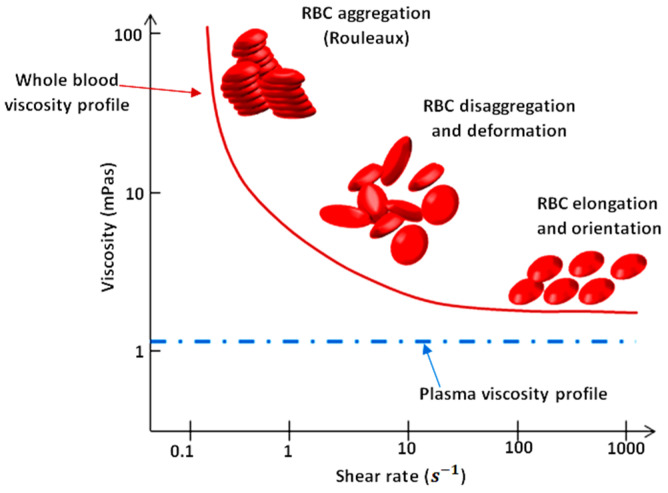
Viscosity profiles of whole blood and plasma. Adapted from Rosencranz [7].

**Figure 2 micromachines-12-00726-f002:**
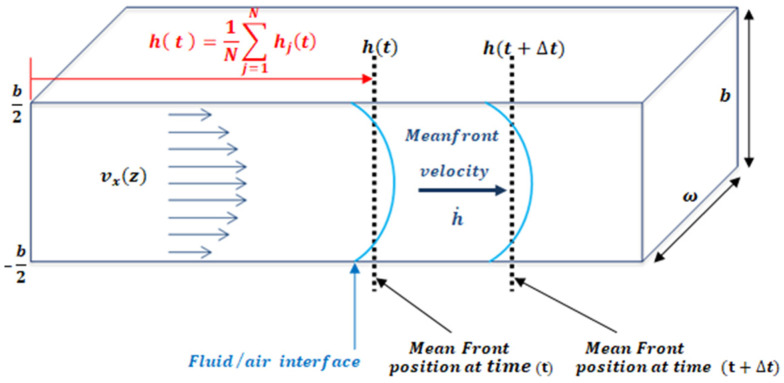
The velocity${\;v}_{x}\left(z\right)$ is the x component of velocity inside the microfluidic channel as a function of height (z); h(t) indicates the mean position of the fluid front. The channel dimensions satisfy the relation b/ω≪1.

**Figure 3 micromachines-12-00726-f003:**
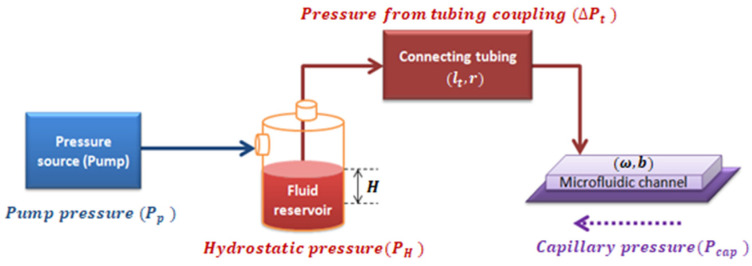
Diagram of the pressures involved in the system.

**Figure 4 micromachines-12-00726-f004:**
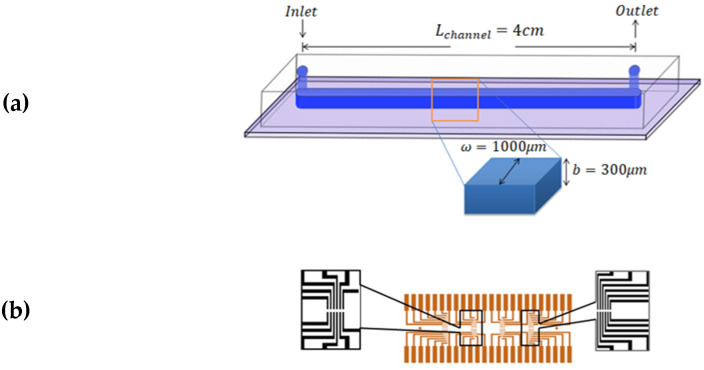
Schematics of the microfluidic consumable. (**a**) The microfluidic channel made of PDMS sealed on a glass substrate. The channel dimensions are ω=1000 μm, b=300 μm, and${\;L}_{c}=4\;\mathrm{cm}.$ (**b**) Gold electrodes printed on the glass surface and beneath the PDMS layer.

**Figure 5 micromachines-12-00726-f005:**
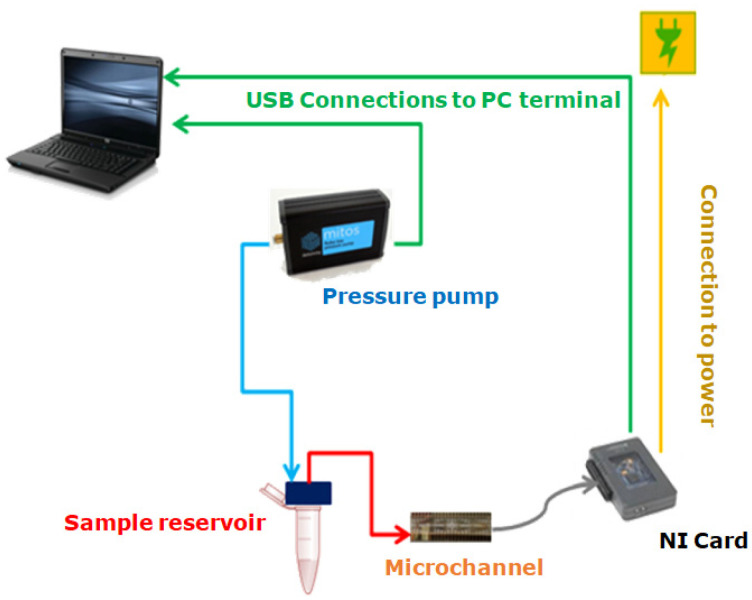
Experimental setup for laboratory use.

**Figure 6 micromachines-12-00726-f006:**
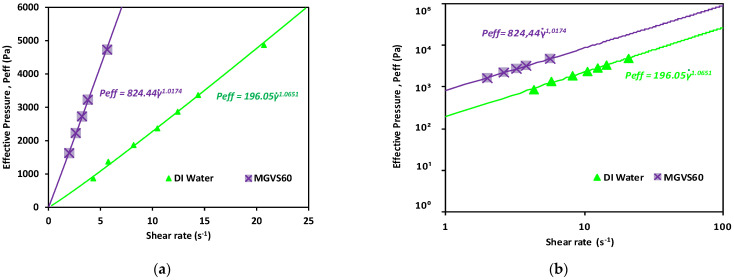
Effective pressure vs. shear rate obtained for the Newtonian calibration fluids studied. Newtonian fluids: DI water and MGVS60: (**a**) linear scale and (**b**) log-log scale.

**Figure 7 micromachines-12-00726-f007:**
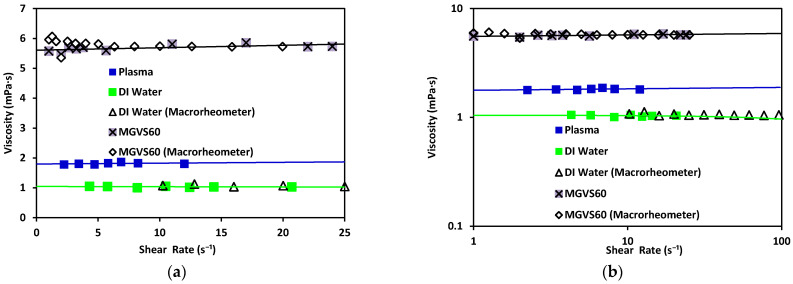
Viscosity vs. shear rate using the microrheometer for DI water (green) and standard MGVS60, and using the macrorheometer: (**a**) linear scale and (**b**) log–log scale.

**Figure 8 micromachines-12-00726-f008:**
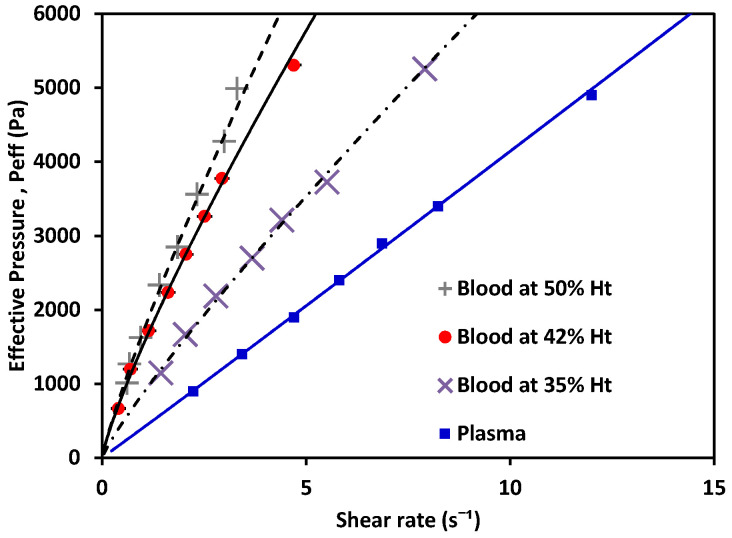
Effective pressure vs. shear rate of blood, a non-Newtonian fluid, at 50% Ht, 42% Ht, 35% Ht; and plasma, a Newtonian fluid.

**Figure 9 micromachines-12-00726-f009:**
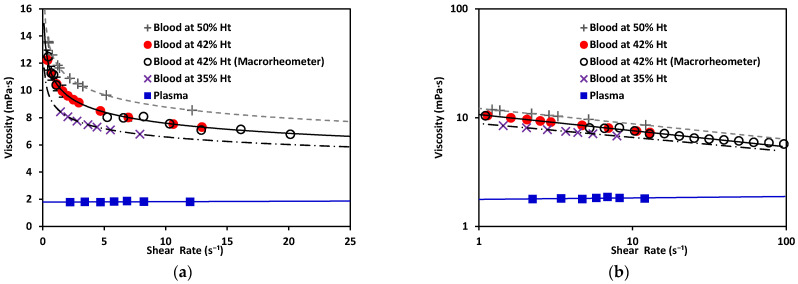
(**a**) Viscosity vs. shear rate of blood, a non-Newtonian fluid, at 50% Ht, 42% Ht, and 35% Ht. Blood at 42% Ht using the macrorheometer (white circles). Viscosity vs. shear rate of plasma, a Newtonian fluid. (**b**) Viscosity vs. shear rate of non-Newtonian and Newtonian fluids in log–log scale.

**Figure 10 micromachines-12-00726-f010:**
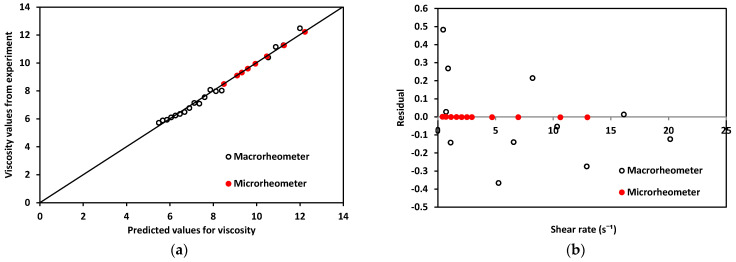
(**a**). Experimentally obtained viscosity values vs. predicted values for both devices using blood. (**b**) Residuals for macro and microrheometers.

**Table 1 micromachines-12-00726-t001:** Characteristic values for water, MGVS60 plasma, and blood.

Sample	$\rho \left(\frac{kg}{m^{3}}\right)$ at 25 °C	$P_{H}\left(Pa\right)$ at H = 5 mm	$\tau \left(\frac{N}{m}\right)$ at 25 °C	θ(º)
Water	1000	49.00	0.072	102.42
MGVS60	1134	55.56	0.107	106.90
Plasma	1025	50.22	0.057	102.27
Blood	1050	51.45	0.058	* (Table 2)

* Contact angle for blood was nonconstant at different pressures. See Table 2.

**Table 2 micromachines-12-00726-t002:** Contact angle and capillary pressure at each pump pressure applied for healthy blood at 42% hematocrit and RT (25 °C). $P_{\mathrm{cap}}$ was obtained using Equation (A20).

$P_{p}\left(Pa\right)$	θ(°)	$P_{cap}\left(Pa\right)$
500	103.24	115.11
1000	106.94	146.46
1500	109.60	168.62
2000	111.76	186.35
2500	113.62	201.37
3000	115.27	214.54
3500	116.76	226.35
5000	120.64	256.22

**Table 3 micromachines-12-00726-t003:** Viscosity obtained using the developed microrheometer (with the corresponding *n* values obtained) and a commercial macrorheometer and electronic detection methods. The samples analyzed were DI water and MGVS60, a commercial standard fluid.

Sample	Viscosity (mPa·s) Microrheometer	n	Viscosity (mPa·s) Macrorheometer
DI water	1.08 ± 0.03	1.017	1.02 ± 0.01
MGVS60	5.87 ± 0.05	1.065	5.88 ± 0.02

**Table 4 micromachines-12-00726-t004:** Values for prefactor *m* and *n* for Newtonian and non-Newtonian fluids.

Sample		*m* (Pa·s)	*n*
*Blood at 50% Ht	0.0122		0.8500
*Blood at 42% Ht	0.0107		0.8519
*Blood at 35% Ht	0.0088		0.8720
MGVS60	0.0058		1.0174 ± 0.002
Plasma	0.0018 ± 0.0002		1.0131 ± 0.028
Water	0.0010 ± 00101		1.0651 ± 0651

*Blood viscosity values presented at $\dot{\gamma }=1$.

## Data Availability

Data is contained within the article.

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
