# Peer review of "Microrheometer for Biofluidic Analysis: Electronic Detection of the Fluid-Front Advancement"

_micromachines, 2021, doi:10.3390/mi12060726_

Round 1
Reviewer 1 Report
In the manuscript, the authors suggested a viscosity measurement method using a fluidic system including pressure source, a circular tubing, and a microfluidic system. Metal electrodes were installed to detect air-liquid interface in microfluidic device. Authors derived analytical expression of overall pressures, and proofed its effectiveness using experimental demonstrations. After reading the manuscript carefully, the reviewer request for authors to update following several issues for improving the manuscript as current form.
1. It was not clear for novelty of the current manuscript. For example, the authors suggested a detection method of fluid advancement in microfluidic device with metal electrodes installed over a single microfluidic channel. In other words, please discuss method and advantage in details, when compared with previous studies.
2. When deriving analytical formula of shear rate in microfluidic channel, it was assumed that fluid flows as 2D model instead of 3D model. In this study, the channel had 300 um depth and 1000 um width. Is it reasonable for assuming 2D flows for the channel ?
3. In the manuscript, was ‘microfluidic channel’ used to detect fluid advancement ? Throughout fluidic system, two representative geometries (i.e., circular tubing and rectangular channel) were used to measure fluid viscosity. However, when non-Newtonian fluid flows in the two geometries, viscosity varies depending on a tubing and a rectangular microfluidic channel. According to manuscript, viscosity was calculated at the tube. But, shear rate was calculated at the microfluidic channel. Thus, there is still discrepancy while calculating viscosity and shear rate. Please modify or correct it appropriately.
4. In the abstract, it is necessary to add ‘motivation’ in the first line and ‘results at the last line’.
5. In the conclusion, it had been written down so long. Please rewrite it shortly.
6. The title is written so general. The word ‘electro’ lead to mislead the manuscript. Thus, it is necessary to modify ‘..electro..’ concretely.
Reviewer 2 Report
This paper introduces a new technique based on microfluidic technology to quantify the viscosity of red blood cells. It uses the so-called "front velocity" to determine the shear rate value. The authors have also presented a mathematical model to explain the viscosity of the Newtonian and Non-newtonian (shear thinning) fluids as a function of shear rate. they have studied water and plasma as the Newtonian fluids and red blood cells as the Non-newtonian system.
The presented results are convincing and the approach is interesting, however, the paper contains grammatical errors and requires significant English improvement.
Author Response
We have done further English proofreading in order to improve the quality of the manuscript. Corrections regarding grammar and punctuation have been done. Style has also been addressed to add clarity.
Reviewer 3 Report
comments in annex file.

Round 2
Reviewer 1 Report
Although the authors tried to reply the issues raised the reviewer, the following issues were still not resolved. For the reason, the revised manuscript as current form should be improved substantially for consideration of publication.
- It was not clear for novelty of the current manuscript. According to previous papers (Deepark E. Solmon et al, Rheol Acta, 2017: Niminsha Srivastava et al, Analytical Chemistry, 2006), the method which the authors suggested was already demonstrated while measuring blood viscosity. Thus, the reviewer wants to know the substantial contributions of the suggested method. Please discuss method and advantage in details, when compared with previous studies.
- When deriving analytical formula of shear rate in microfluidic channel, it was assumed that fluid flows as 2D model instead of 3D model. In this study, the channel had 300 um depth and 1000 um width. Is it reasonable for assuming 2D flows for the channel? In other words, the reviewer want to see quantitative data while varying aspect ratio, especially by means of numerical/analytical/ or experimental approaches.
- In the manuscript, two representative geometries (i.e., circular tubing and rectangular channel) were used to measure fluid viscosity. However, when non-Newtonian blood flows in the two geometries, viscosity varies depending on a tubing and a rectangular microfluidic channel. According to manuscript, viscosity was calculated at the circular tubing. But, shear rate of blood flow was calculated at the microfluidic channel (i.e., Eqn (1)) as rough expression. Thus, there is still discrepancy while calculating viscosity and shear rate. Please modify or correct it appropriately.
Reviewer 3 Report
Minor Revisions in the attached file.

Round 3
Reviewer 1 Report
As the authors discussed the issues the reviewer raised appropriately,
the reviewer recommend publication of the manuscript as current form.